# Botulinum Toxin Injection for Medically Refractory Neurogenic Bladder in Children: A Systematic Review

**DOI:** 10.3390/toxins13070447

**Published:** 2021-06-28

**Authors:** Shu-Yu Wu, Shang-Jen Chang, Stephen Shei-Dei Yang, Chun-Kai Hsu

**Affiliations:** Division of Urology, Taipei Tzu Chi Hospital, Buddhist Tzu Chi Medical Foundation, New Taipei City 23142, Taiwan; nobookrain2014@gmail.com (S.-Y.W.); krissygnet@gmail.com (S.-J.C.); urolyang@gmail.com (S.S.-D.Y.)

**Keywords:** botulinum toxin, neurogenic bladder, children

## Abstract

The objective was to evaluate the use of botulinum toxin A (BTX-A) injection in children with medically refractory neurogenic bladder. A systematic review of the literature was conducted using three databases (Medline via PubMed, Cochrane, and EMBASE). Articles evaluating BTX-A in children with neurogenic bladder were collected. The clinical and urodynamic parameters were reviewed for the safety and efficacy evaluation. Sixteen studies were selected into this study and a total of 455 children with medical refractory neurogenic bladder were evaluated. All of the patients had received traditional conservative medications such as antimuscarinics and intermittent catheterization as previous treatment. The duration of treatments ranged from 2 months to 5.7 years. Improvements in incontinence and vesicoureteral reflux were the most common clinical outcomes. The detrusor pressure, bladder capacity and bladder compliance improvement were the most common urodynamic parameters which had been reported. However, patient satisfaction with the procedure remained controversial. There was only a minimal risk of minor adverse effects. In all of the studies, BTX-A injection was well tolerated. In conclusion, BTX-A injection appears to be a safe and effective treatment in the management of medically unresponsive neurogenic bladder in children. There is currently no evidence that the use of BTX-A injection could be used as a first-line therapy for neurogenic bladder in children.

## 1. Introduction

Neurogenic bladder is a complex disease which often seriously affects daily activities. It often presents as detrusor overactivity (DO), urinary leakage and hydronephrosis. DO results in decreased bladder capacity, low compliance and high intravesical pressure. These may result in cortical thinning, febrile urinary tract infection (UTI) and even loos of renal function [1,2]. Children with neurogenic bladder are at an increased risk of UTI and renal function deterioration.

Routine management includes oral medications such as anticholinergics or β-3 agonists, timely clean intermittent catheterization (CIC) or long-term indwelling catheterization [3,4]. The goal is to reduce intravesical pressure to protect renal function. However, if the pressure remains high or lower urinary tract symptoms persist, intravesical botulinum toxin A (BTX-A) injection is considered as alternative to irreversible surgical interventions. Current studies have demonstrated improvements in both objective symptoms and subjective parameters by intravesical BTX-A injection. BTX-A treatment may avoid or delay complex reconstructive surgery [5,6,7,8]. However, there is little evidence on the effects of BTX-A on pediatric neurogenic bladder patients. Therefore, the aim of this study was to perform a systematic review of intravesical BTX-A injection on treating children with refractory neurogenic voiding dysfunction.

## 2. Materials and Methods

We used the Preferred Reporting Items for Systematic Reviews and Meta-Analyses (PRISMA) guidelines to report this systematic review [9]. We did not register a protocol for this systematic review. Three databases (Medline via PubMed, Cochrane, and EMBASE) were searched for qualified studies. The following keywords were used, along with Boolean operators: “neurogenic”, “bladder”, and “botulinum toxin”. The following search strategy was used on Medline via PubMed: (Botox, “neurogenic bladder”) and (botulinum toxin, “neurogenic bladder”). The search was conducted on 1 February 2021. Only articles published in English which included patients aged under 18 years were collected.

After removing duplicates, the titles and abstracts of the studies were reviewed by two independent reviewers (Hsu and Wu). If the article is selected only by one reviewer, it will be retained first and analyzed in the next step. Case reports, current review articles, commentaries, editorials, letters and abstracts were excluded. Studies focusing on adult patients and those that included children along with others but did not report the outcomes of each subgroup separately were also excluded. Then the full texts were obtained and reviewed by all the authors. For any other related research, references were also checked manually. Studies focusing on non-neurogenic bladder were excluded. For each article, the following parameters were collected: type of study; patient numbers and genders; mean age of the patients; neurological disease(s); BTX-A treatment modality and duration; and clinical and urodynamic parameters for the safety and efficacy evaluation.

## 3. Results

A total of 30 studies were screened, of which 16 were included according to the PRISMA protocols (Figure A1). Of these 16 studies, five were prospective without placebo control. Only two randomized control trials were identified. A total of 455 patients with neurogenic bladder were evaluated, ranging from seven to 65 patients per study. The most common neurologic deficit was meningomyelocele. The patients with a poor response to clean intermittent catheterization or medical treatment were treated with BTX-A injections into the detrusor of urinary bladder or the external sphincter. The age of the patients varied widely in these patients. The dosage of BTX-A ranged from 100 U to 1200 U, most studies use the onabotulinum toxin (100–500 IU, 10–50 IU/kg according to the body weight); only two studies use the onabobotulinum toxin (500–1200 IU, 40 U/kg according to the body weight). The number of injection sites range from 20 to 50. Ten studies showed the results of multiple injections. Four studies applied reinjection after the urodynamics parameters had returned to baseline or in cases of persistent or recurrent incontinence. All of the studies used an interval of at least 3 months between each injection, with the interval ranging from months to years. There was a trend of a stable urodynamic response after multiple injections. All of the studies reported a low adverse event rate and that the injection was safe. The clinical demographic data are shown in Table 1.

### 3.1. Clinical Parameters

Among the 16 studies, 15 of them report the improvement of clinical parameters. The clinical parameters included incontinence, vesicoureteral reflux (VUR), UTI, bowel dysfunction, hydronephrosis and patient reported satisfaction. The most commonly reported symptom was urinary incontinence, and 13 studies reported an improvement in incontinence, with a post-injection improvement rate of 54–100%. Moreover, 30–100% of the patients were totally continent after the injection. The second most commonly reported symptom was VUR, and six studies reported improvements in VUR. Of these studies, only one reported no change in VUR after the injection [15]. The other studies reported improving or improved reflux, with a rate of improvement ranging from 73–100%. Other clinical parameters including hydronephrosis, fecal continence and UTI were rarely reported. Three studies reported an improvement in hydronephrosis [11,19,20].

In these studies, some of the patients had improvements in hydronephrosis but others did not, although the number of patients was limited (3–10 patients in each study). The improvements in hydronephrosis were reported to be related to resolving reflux. Bowel dysfunction was only reported in two studies [10,12], and the improvement rate ranged from 46–93%. One study reported improvements in UTI [10], in which 26 children with a previous UTI did not have an episode during four months of follow-up after BTX-A injection. The patient’s reported response was also an important clinical parameter. Two studies reported opposite results of patient satisfaction and changes in urodynamics [13,18]. The only study that mentioned urethral injections was a prospective, randomized, double-blind control trial by Safari et al. They reported that BTX-A injections in both the sphincter and detrusor seemed to be more beneficial in decreasing postvoiding residual volume and reducing symptoms compared with an intradetrusor injection alone even unable to facilitate spontaneous emptying [12].

### 3.2. Urodynamic Parameters

Urodynamic changes were also reported in the studies. Although the items emphasized in each article were different, they mainly focused on detrusor pressure, bladder capacity and bladder compliance. Ten studies reported baseline urodynamic DO according to urodynamic studies before the injection. Eleven studies reported a significant decrease in maximum detrusor pressure (Pdet). The patients in these studies had a very high Pdet at baseline (43–139 cm H_2_O), which then decreased to 22–83 cm H_2_O after the injection. In addition, bladder pressure was reported to be reduced by up to 50%. Fourteen studies reported enlarged bladder capacity after BTX-A injection, ranging from 1.45 to 3.64 times (40–217 mL) compared to baseline. Bladder compliance was also significantly improved in these studies. Among the seven prospective studies, bladder compliance was reported in three [5,11,13]. Deshpande et al. reported that the effect on bladder compliance was most pronounced 3 months after the injection. They reported that bladder compliance improved from 4.42 mL/cm H_2_O at baseline, to 9.33 mL/cm H_2_O at 3 months, and then the effect progressively decreased (8.12 and 6.35 mL/cm H_2_O at 6 and 9 months after the injection, respectively) [13]. Altaweel et al. reported significant improvements in bladder compliance after both the first injection (from 5.2 to 13 mL/cm H_2_O) and second injection (from 6 to 15.1 mL/cm H_2_O) [11]. Figueroa et al. [5]. reported the effect of repeat injections. Compared with baseline, after the first injection, the mean bladder compliance improved by 45.2%; after subsequent injections, the value increased by 55.1%. In Neel et al.’s prospective randomized control trial, continuous oxybutynin treatment after BTX-A injection did not provide extra benefits based on the urodynamic parameters (Pdet, bladder capacity) [23]. In addition to the general parameters mentioned above, Marte’s study focused on leak point volume (LPV), leak point pressure (LPP), and the specific volume at 20 cm H_2_O pressure (SC 20) [17]. They reported a significant average increase in LPV (66.45%) and in SC 20 (118.57%) after surgery. However, the difference between preoperative and postoperative LPP was not significant. Two studies also investigated the reflex volume [9,18] and reported improvements similar to bladder capacity, although the statistical difference remained controversial. In the 11 studies which reported the effect of repeat injections, there were two important findings. First, the effect of BTX-A decreased with time, usually after 6–9 months. However, repeat injections after symptom relapse were still as beneficial as the first injection. Second, in the patients in whom the first BTX-A injection was not as effective as expected, repeat injections provided better results.

### 3.3. Safety and Side Effects

BTX-A injections appeared to be safe, as only a few patients had adverse events. Across all retrieved studies, six studies mentioned adverse effects of BTX-A injections. The most common adverse effect was temporary hematuria, and one study reported a high rate (80.9%) of post-injection hematuria [17]. However, another study showed a much lower rate of hematuria (10.9%) [20]. Post-injection UTI were the other reported adverse effects. Five studies reported the rates of UTI after each injection (14.3%, 4.3%, 2.7%, 2.1% and 3.3% respectively) [13,17,18,21,22]. The other studies reported mild or even no side effects.

## 4. Discussion

Neurogenic bladder presents with various bladder dysfunctions, including detrusor-sphincter dyssynergia, DO and poor compliance. Failure of control will lead to recurrent UTI and upper urinary tract deterioration and loss of renal function. Urinary symptoms are present in nearly all of these patients, in whom renal failure and UTI may cause death [24,25]. Preservation of renal function and elimination of incontinence are crucial in bladder management while improving quality of life.

Oral antimuscarinics often act as first line therapy for detrusor overactivity or poor compliance. Many antimuscarinics are currently available worldwide. However, oxybutynin is the only agent approved by the FDA on children. The effectiveness of treatment is well known as well as its adverse event including dry eyes, dry skin, constipation and stomach-intestine disorders. Moreover, it can also cause disturbances require dose adjustment or even discontinuation of oxybutynin treatment. Recently, a new beta-3 agonist agent has become available as add-on therapy or alternative. If pharmacological treatments fail, surgery such as bladder augmentation will be carried out as final solution. However, those complex surgical procedures yield long-term serious complications such as stenosis, stone formation, infection or metabolic disturbance [26,27].

BTX-A is a 2-chain peptide neurotoxin protein produced by the Gram-positive, anaerobic bacterium *Clostridium botulinum*. Physiologic activation starts by proteolytically break down into the 100 kDa heavy chain and 50 kDa light chain. The light chain of the botulinum neurotoxin binds with high specificity to the glycoprotein on cholinergic nerve terminals. Ultimately, BTX-A causes acetylcholine-containing vesicles unable to reach the nerve terminal. This blockage results in muscle paralysis and relax of urinary bladder [3,28]

Application of BTX-A in neurourologic field has increased in past two decades. Schurch et al. first reported efficacy of intradetrusor injections of BTX-A in 2000 [25]. After a series of trials, BTX-A injection have become the standard of care for neurogenic detrusor overactivity refractory to antimuscarinics in adults with spinal cord injury or multiple sclerosis. However, in pediatric neurogenic bladder, BTX-A is applied as substitute to anticholinergics when adverse events such as urinary incontinence or upper urinary tract dilation persisted. If BTX-A does not work, reconstructive surgery of the lower urinary tract is final resort to preserve renal function [29]. In addition, BTX-A injections could be used as a diagnostic tool for the evaluation of neurogenic incontinence caused by bladder outlet incompetence.

In the studies, intravesical BTX-A injections improved both clinical symptoms and cystometric parameters in children who got neurogenic detrusor overactivity (NDO) [5,6,7,10]. The first report of the use of BTX-A injection in children was published by Schulte-Baukloh in 2002 [9]. Subsequently, several studies have reported that BTA injections can improve both urodynamic parameters (reflex volume, bladder capacity, maximal detrusor pressure and bladder compliance) and symptoms of NDO [5,9,28,30]. The clinical outcomes were favorable with a completely resolved urinary incontinence in one third to nearly 100% of the patients. In addition, there was high percentage of clinical incontinence resolution, although the degree of improvement varies between each study. Post-injection UTI were the most common adverse events. Other significant adverse events which are commonly noted in adults such as increasing post-void residual urine, acute urine retention, and general muscle weakness were rarely mentioned.

The International Children’s Continence Society (ICCS) does not define which urodynamic outcome parameters should be evaluated. Therefore, bladder compliance, maximum detrusor pressure (MDP) and maximum cystometric capacity were most commonly used for outcome analysis. An increase in maximum cystometric capacity and decrease in MDP were reported in nearly all of the studies. However, in some series, the MDP was still above 40 cm-H2O after BTX-A injection. The increase in bladder compliance was also statistically significant. Bladder augmentation or other reconstructive surgery accompany with perioperative risk and long-term adverse event. To avoid or at least delay major operation is the ultimate goal of oral pharmacologic treatment and intradetrusor injection in pediatric neurogenic patients. Most of the enrolled studies did not report the number of patients who end up with bladder augmentation despite BTX-A injections. Furthermore, the studies did not report in how many major reconstructive surgery was delayed during follow-up, and it was not clear if these patients all had a low-compliance bladder. Still, effort on avoiding irreversible major operation is meaningful for pediatric patients because of complications and possible cancer development decades after augmentation. Interesting, the most important renal function deterioration was not mentioned in all literatures. We have many studies focus on the improvement of VUR and hydronephrosis, but not the exact renal function by the blood test or other examinations. It might due to the difficulty for children to draw blood and the lack of long-term research.

Although forming neutralizing antibodies may downgrade treatment effect after repeated injection of BTX-A. The problem has seldom been reported before, and the levels of antibodies did not increase after repeat injections [31]. BTX-A injections may lead to axonal sprouting and the generation of new synaptic contacts on paralyzed muscle fibers. The problem was reported in striated muscle rather than smooth muscle of urinary bladder. No structural effects have been shown on the effects of BTX-A injections on the human detrusor. Recurrent puncture during repeated injections of BTX-A may also result in scarring and subsequent reduction of bladder compliance [32]. Despite the concerns, repeated BTX-A injection still showed similar improvement of urodynamic parameters in each intervention. The lasting effect varies from 8 to 15 months in enrolled studies.

Besides traditional intradetrusor BTX-A injections, there are also other methods of delivery such as electromotive drug application (EMDA) and liposomal drug delivery. EMDA uses iontophoresis and electrophoresis to deliver the pharmacological agents into deeper tissue layers. Two electrodes will be used to provide electric current during the operation. EMDA has been used widely in delivering different medication into human bladder wall in adult urology [33]. Kajbafzadeh et al. have published two studies about BTX-A delivery use EMDA in 2011 [34,35]. Through immunohistochemical staining, they showed that EMDA provided more BTX-A penetration in bladder suburothelial layers compare to the traditional injection. In their trial, EMDA got significant improvement in maximum cystometric capacity, mean reflex volume and mean end-fill pressure. In a more recent study in 2019, Koh et al. reported that BTX-A/EMDA got poor performance when compared with conventional intravesical BTX-A injections. The negative finding may imply that large size of the BTX-A molecule is hard to penetrate the thick and hypertrophy bladder wall in NDO [36]. Liposomes are another theoretically ideal carrier in transurothelial drug delivery. It could penetrate the urothelial barrier through endocytosis. The recent clinical studies showed that BTX-A encapsulated within liposomes (Lipo-BTX) may be useful in adult patients with a non-neurogenic overactive bladder [37,38,39]. It provided benefit on frequency and urgency but not on urinary incontinence or residual urine volume. Therefore, Lipo-BTX could affect the afferent signaling of the urothelium but not work on the efferent motor activity of the detrusor. Although Lipo-BTX may be effective for the storage symptoms of an overactive bladder, the application on children with refractory neurogenic bladder remain uncertain due to minimal effects on decreasing detrusor contraction.

BTX-A injection on neurogenic bladder children result in good outcomes in clinical or urodynamic parameters in most of the studies. Although antimuscarinics was applied as first line treatment, threat of no response and/or side effects persisted. BTX-A injection was proposed to be a favorable second-line intervention after antimuscarinics in refractory neurogenic bladder. The safety of BTX-A injection has previously been reported in the literature [17,18,21,22]. In these studies, the rate of adverse events ranged from 2–5%. Constipation, fatigue, headache, and hypertension are most common adverse events. After all, BTX-A injection is favorable safe in all of the included studies.

## 5. Conclusions

BTX-A injections for refractory neurogenic bladder are effective in resolving both urinary incontinence and improving urodynamic parameters in most children. To maintain the effect, multiple injections are often needed. General anesthesia is routinely carried out in children for BTX-A injections. Alternative route has been investigated. However, there is no definite result in pediatric setting. From enrolled studies, BTX-A may injection is a favorable intervention in avoidance of major reconstructive surgery such as bladder augmentation. Our results provide further evidence that intradetrusor BTX-A injection is a safe and effective option in the management of children with neurogenic bladder who fail to respond to conservative medical therapy.

## Figures and Tables

**Table 1 toxins-13-00447-t001:** Summary of studies evaluating the use of botulinum toxin A in children with neurogenic bladder (patients, treatment, and response characteristics).

First Author	Type of Study	Number of Patients (*n*, M/F)	Mean Age (Year)	Neurologic Diseases (*n*)	Previous Treatment Modalities	Treatment Modality	Treatment Duration	Follow Up Period	Clinical Parameters	Urodynamic Parameters	Adverse Event (*n*)	Primary and Special Outcomes
Heinrich Schulte-Baukloh [8]	Retrospective chart analysis	10, 6/4	11.2	MMC 8 Intraspinal astrocytoma 1 SCI 1	anticholinergic medications and CIC	12 IU/kg, diluted in 15 to 20 mL normal saline, injected into 30 to 50 sites	repeated injection (at least three injections; 4 had received five or more injections)	1, 3, and 6 months	NR	Reflex volume, Pdetmax, maximun bladder capacity, bladder compliance	NR	After the first versus the fifth injection reflex volume increased by 81% versus 88%, maximal detrusor pressure decreased by 7% versus 39% maximal cystometric bladder capacity increased by 88% versus 72%
Abdol-Mohammad Kajbafzadeh [10]	Single-center, prospective, unrandomized study	26, 20/6	6.9	MMC	anticholinergic medications and CIC	10 IU/kg, diluted in 20 mL of normal saline, injected intravesically into at least 40 sites	single injection	4 months	Incontinence, VUR, urinary tract infection, fecal continence	Pdetmax, maximun bladder capacity	no major treatment related event	73% complete dry, 88% total improvement in urine incontinence; significant improvements in mean maximal detrusor pressure and average maximal bladder capacity
W. Altaweel [11]	Prospective study	20, 8/12	13	MMC	anticholinergic medications and CIC	5 IU/kg (maximum 300 IU) diluted 10 times in normal saline, injected to give 10 IU per site	repeated injection (reassess approximately 3 months after each treatment, repeat injection if incontinent)	3 months after each treatment (total mean follow-up 17.2 months)	Incontinence, hydronephrosis	Pdetmax, bladder compliance	no treatment related event	65% continent; significant improvements in mean bladder capacity, maximum detrusor pressure and compliance
Saeed Safari [12]	Prospective, single center, double-blind, randomized control trial	Group A (30, 13/17) Group B (30, 14/16)	Group A (6.58) Group B (6.71)	MMC	anticholinergic medications and CIC	Group A (10 U/kg, diluted in 20 mL of normal saline, injected into 40 points of bladder); Group B (8 U/kg, injected in the same way as in group A and the remaining (2 U/kg) was injected by 4 additional injections in external urethral sphincter)	single injection	3 and 6 months	Incontinence episodes, Constipation, Vesicoureteral reflux grade	Detrusor-sphincter dyssinergia, maximum detrusor pressure, post-voiding residual volumn, bladder capacity	NR	BTX-A injections in both sphincter and detrusor seems to have extra benefits in voiding
Aniruddh V Deshpande [13]	Prospective, non-randomized study	7, 6/1	16	Spinal bifida	CIC and oxybutynin (dose range 5 mg b.d.–5 mg q.i.d.)	10 IU/kg, (maximum 300 IU) diluted as 10 units per mL in saline, injected into the detrusor at approximately 20 to 30 sites	single injection	1, 3–6 and 9 months	Incontinence, satisfaction score	Bladder capacity, bladder compliance	UTI 1	significant improvements in bladder compliance and incontinence
R Le Nué [14]	Retrospective chart analysis	8, 3/5	12.4	SCI 6 Stroke 2	maximal oral anticholinergic treatment, CIC	12 IU/kg (maximal 300 IU), diluting to a concentration of 100 IU/10 mL before 2008 and then 100 IU/5 mL of 0.9% saline, injected into 10 to 30 sites	repeated injection (2–6 injections, repeat injections depended on the initial urodynamic status)	6 months after each treatment (total mean follow-up 47 months)	continence score	Pdetmax, bladder compliance, maximun bladder capacity, safe capacity	NR	Improvements in the mentioned parameters
Maya Horst [15]	Retrospective chart analysis	11, 1/10	6.7	MMC	anticholinergic medications	10 IU/kg (maximal 300 IU), diluting to a concentration of 100 IU/10 mL of 0.9% saline, each injection contained 0.3–0.5 mL	repeated injection (1–4 injections, reinjection was performed if the urodynamic follow up study showed compliance and pressure returned to baseline values)	3 and 12 months after each injection	VUR	Changes in bladder compliance, maximal bladder capacity, maximal detrusor pressure	NR	detrusor pressure decreased by 17% and bladder capacity increased by 33%; similar effect on capacity and detrusor pressure could be achieved with repeated injection
Pawel Kroll [16]	prospective, non-placebo-controlled study	65, 34/31	6.7	MMC 61 Sacral agenesis 3 Cerebral palsy 1	CIC and oral oxybutynine	50 IU/kg (maximal 500 IU), diluted in 10 mL normal saline, injected each with 0.5 mL of the solution	single injection	6 and 12 months	continence	Maximun catheterized volume, maximun volume of leak point pressure	NR	Improvements in the mentioned parameters
Antonio Marte [17]	Retrospective chart analysis	47, 25/22	10.7	MMC	anticholinergic medications and CIC	200 IU, diluted in 20 mL of 0.9% saline solution, not exceeding the dosage of 12 IU/kg, injected for a total of 20 injections	repeated injection (1–3 injections, for the recurrence of symptoms)	6, 12 and 24 weeks after each injection (total mean followup 5.7 years)	Incontinence, VUR	Mean leak point volume, mean leak point pressure, specific capacity at 20 cm H_2_O, bladder capacity	slight hematuria 38, UTI 2, gastric pain 2, facial flushing 2, mild hyposthenia of the lower limbs 5	significant 66.45% average increase of leak point volume, significant 118.57% average increase of specific bladder capacity at 20 cm H_2_O
V. Figueroa [5]	Prospective study	17	10.7	spina bifida/spinal dysraphism	anticholinergic medications	10 IU/kg (maximal 200 IU), diluted in normal saline to a concentration of 10 units/cc	repeated injection (average of 2.5 injections/patient; range, 1–6)	3, 6 months after each injection (total mean follow-up 4 years)	Incontinence	Mean bladder capacity, detrusor compliance, bladder volume, bladder compliance	NR	The optimal response occurs with a maximum administration of BTX-A up to 300 units.
Sang Woon Kim [18]	Retrospective chart analysis	37, 22/15	7.49	Spina bifida 29 Syrinx 1 Cerebral palsy 4 Guillain-Barre syndrome 1 Spinal cord hemangioma 1 Post meningitis sequelae 1	high-dose anticholinergic medications	10 IU/kg (maximal 200 IU), diluting to a concentration of 100 IU/5 mL of 0.9% saline, injected into 20 to 40 sites	single injection	1, 3, 6 months	Patient Global Impression of Improvement (PGI-I)	detrusor-sphincter dyssynergia, maximum cystometric capacity, estimated bladder capacity, Residual urine volume, maximal detrusor pressure, reflex detrusor volume, bladder compliance, open bladder neck	UTI 1	preoperative bladder compliance and open bladder neck were important predictors
M. K. Khan [19]	Retrospective chart analysis	22, 16/6	10	Myelomeningocele 10 Anorectal malformation 3 Spinal cord trauma 3 Tethered cord syndrome 2 Caudal regression syndrome 2 Sacrococcygeal teratoma 1 Transverse myelitis 1	anticholinergic medications	10 IU/kg (maximal 300 IU), diluted in normal saline to a concentration of 10 units/cc	repeated injection (Four patients have received two or more injections for the recurrence of symptoms)	3, 6 months after each injection (total mean follow-up 11 months, range 3–38)	Incontinence, hydronephrosis	Cystometric bladder capacity, mean maximum detrusor pressure	NR	patients with anticholinergics intolerance seen to have be more effective after BTX-A injection than those with anticholinergic refractory
Cagri Akin Sekerci [20]	Retrospective chart analysis	19, 4/15	10.3	myelodysplasia	anticholinergic medications	10 IU/kg (maximal 200 IU), diluted in normal saline to a concentration of 10 units/cc, injected to give 10 IU per site	repeated injection (1–5 injections, repeated only after the relapse of incontinence accompanying deterioration in urodynamic findings)	every 3 months after each injection	Incontinence, VUR, hydronephrosis	Maximum cystometric capacity, Maximum detrusor pressure, Compliance	hematuria 2	significant improvements in the mentinal parameters after repeat injection
Juliette Hascoet [21]	Retrospective chart analysis	53, 28/25	8.5	Spina bifida	anticholinergic medications	100–500 U	repeated injection (1–8 injections)	mean follow up of 3.7 years	Incontinence	Resolution of detrusor overactivity, normal bladder compliance, maximum cystometric capacity, maximum detrusor pressure	3 UTI episodes out of 141 injections (2.1%)	66% clinical success rate, 34% urodynamic success rate
Shehryer Naqvi [22]	Retrospective chart analysis	30, 15/15	7.4	MMC 18, Lipomeningocoele 4, Transverse myelitis 1, Sacral agenesis 3	anticholinergic medications and CIC	40 IU/kg, (maximal 1200 IU) diluted in 20 mL sodium chloride solution	repeated injection (Median number of injections was 3 (range 2 to 5)	median 2 months (range 0–29)	incontinence, VUR	Bladder compliance, cystometric capacity, maximum neurogenic detrusor overactivity	abdominal pain 1, UTI 1	significantly improved cystometric capacity and maximum neurogenic detrusor overactivity, no significant difference in urodynamic parameters between first and last injections
Khalid Fouda Neel [23]	Prospective, single center, randomized trial	Group 1: 12 9/3 Group 2: 11 6/7	Group 1: 6.1, Group 2: 5.1	MMC	oxybutynin and CIC	12 IU/kg (maximal 300 IU) Group 1: continued to receive anticholinergics; Group 2: dicontinued to receive anticholinergics	single injection	12 months	incontinence	Maximum cystometric capacity, maximum detrusor pressure	NR	Oxybutynin did not have augmentative effect after BTX-A injection

CIC: clean intermittent catheterization, BTX-A: botulinum toxin A, MMC: myelomeningocele, NR: not reported, Pdetmax: detrusor pressure at maximum flow rate, SCI: spinal cord injury, UTI: urinary tract infection, VUR: vesicoureteral reflux.

## Data Availability

Data available in a publicly accessible repository.

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
