# Peer review of "Botulinum Toxin Injection for Medically Refractory Neurogenic Bladder in Children: A Systematic Review"

_toxins, 2021, doi:10.3390/toxins13070447_

Round 1

Reviewer 1 Report

Well written paper and its strength lies in the meta analysis nature. However, the conclusions are well accepted in the clinical practice already. 

Author Response

Thanks for the comments. Botox injection was widely used in adult urology, but still lack of large trial in children. After review of the literature, botulinum toxin type a injection is a safe and effective option in the management of children with refractory neurogenic bladder before irreversible operation.

Reviewer 2 Report

This is an important topic.
Young people must be accurately and

early treated to prevent kidney failure.

There are still not present in Literature 

papers that present a sistematic

review on BTX in children, cause most of

papers refer to specific sub-populations.

I consider this a valid contribute 

Author Response

Response to Reviewer 2 Comments

This is an important topic. Young people must be accurately and early treated to prevent kidney failure. There are still not present in literature. Papers that present a systematic review on BTX in children, cause most of papers refer to specific sub-populations. I consider this a valid contribute.

Reply: Thanks for the comments. Most papers focused on the clinical parameters such as incontinence, vesicoureteral reflux, urinary tract infection and hydronephrosis. Some studies had mentioned about the improving of vesicoureteral reflux and hydronephrosis after BTX-A injection. But there’s no parameters focus on the exact renal function. We have added the statements in discussion (Line 207-211).

Reviewer 3 Report

1/16 …  Conclusion: BTX-A injection appears to be asafe and effective treatment in the management  a safe

2/71 …The dosage of BTX-A ranged from 100 U to 1200 U: It is imperative that you emphasize that different formulations have been used (onabotulinum toxin and abobotulinum toxin) as the units required are vastly different.

2/80…All of the studies except one [9] reported improvements in clinical symptoms

3/100: …trial by Safari et al. They reported that BTX-A injections in both the sphincter 100 and detrusor seemed to be more beneficial in decreasing postvoiding residual volume  and 101 reducing symptoms compared with an intradetrusor injection alone[12]. At this point I would mention that spontaneous emptying is unusual in these patients (in order to prevent the reader from concluding that this technique enables spontaneous emptying.

3/127 They reported a significant 66.45% average increase in 127 LPV and a significant 118.57% average increase in SC 20 after surgery more common: 66.5% … and 118.6%

5/215 One factor that can affect the long-term efficacy of BTX-A is the formation of neutral-215 izing antibodies. This has seldom been reported before, and the levels of antibodies did 216 not increase after repeat injections. Cite:  Results of a BoNT/A antibody study in children and adolescents after onabotulinumtoxin A (Botox®) detrusor injection. Schulte-Baukloh H, Herholz J, Bigalke H, Miller K, Knispel HH. Urol Int. 2011;87(4):434-8.

Author Response

Response to Reviewer 3 Comments

1/16 …  Conclusion: BTX-A injection appears to be asafe and effective treatment in the management a safe

Reply: We have corrected the mistake. Thank you. (Line 17)

2/71 …The dosage of BTX-A ranged from 100 U to 1200 U: It is imperative that you emphasize that different formulations have been used (onabotulinum toxin and abobotulinum toxin) as the units required are vastly different.

Reply: Thanks for the comments. We have reviewed the literature for check the dosage. The related description was added to the manuscript. (Line 70-72)

2/80…All of the studies except one [9] reported improvements in clinical symptoms

Reply: We have changed the statement. Thank you. (Line 80)

3/100: …trial by Safari et al. They reported that BTX-A injections in both the sphincter and detrusor seemed to be more beneficial in decreasing postvoiding residual volume and reducing symptoms compared with an intradetrusor injection alone[12]. At this point I would mention that spontaneous emptying is unusual in these patients (in order to prevent the reader from concluding that this technique enables spontaneous emptying.

Reply: Thanks for the comments. We have changed the description. (Line 100-103)

3/127 They reported a significant 66.45% average increase in 127 LPV and a significant 118.57% average increase in SC 20 after surgery more common: 66.5% … and 118.6%

Reply: Thanks for the comments. We have changed the description. (Line 128-129)

5/215 One factor that can affect the long-term efficacy of BTX-A is the formation of neutral-215 izing antibodies. This has seldom been reported before, and the levels of antibodies did 216 not increase after repeat injections. Cite:  Results of a BoNT/A antibody study in children and adolescents after onabotulinumtoxin A (Botox®) detrusor injection. Schulte-Baukloh H, Herholz J, Bigalke H, Miller K, Knispel HH. Urol Int. 2011;87(4):434-8.

Reply: We have adjusted the manuscript and cited the article. Thank you. (Line 212-214; 336-337) 

Round 2

Reviewer 2 Report

I think the manuscript

is well done and satisfies a topic required in the curent Literature